# Genetic Analysis of *HSP70* and *HSF3* Polymorphisms and Their Associations with the Egg Production Traits of Bangladeshi Hilly Chickens

**DOI:** 10.3390/ani14243552

**Published:** 2024-12-10

**Authors:** Md Yousuf Ali, Shakila Faruque, Sadequllah Ahmadi, Takeshi Ohkubo

**Affiliations:** 1Bangladesh Livestock Research Institute, Savar, Dhaka 1341, Bangladesh; shakila_blri@yahoo.com; 2United Graduate School of Agricultural Science, Tokyo University of Agriculture and Technology, Fuchu 183-8538, Japan; sadequllah15@gmail.com (S.A.); takeshi.ohkubo.0533@vc.ibaraki.ac.jp (T.O.); 3College of Agriculture, Ibaraki University, Ami 300-0393, Ibaraki, Japan

**Keywords:** chicken, egg production, HSF3, HSP70, SNPs

## Abstract

Heat stress is a challenging environmental factor that affects poultry production. Adaptation to heat stress is regulated by heat shock proteins (HSPs). In chickens, HSP70 expression is controlled by heat shock factor 3 (HSF3). The genetic diversity of heat shock genes is thought to be closely associated with the productive and reproductive performance of farm animals. The present study aimed to explore the relationships between single-nucleotide polymorphisms (SNPs) in the genes of Bangladeshi hilly chickens and their egg production traits. Sequencing and allele-specific PCR identified two novel SNPs (G-399A and A-68G) in the 5′-flanking regions of HSP70. In addition, three previously identified SNPs in *HSP70* (A258G, C276G and C1431A) and two SNPs in *HSF3* (A-1388G and A-1703G) were also genotyped. A population of 150 breeding hilly chickens was used to analyze the associations between SNPs in these genes and the chickens’ egg production traits. Among the analyzed SNPs, the two novel SNPs (G-399A and A-68G) were significantly associated with egg number and egg weight, respectively. In addition, all other SNPs except A-1703G were also found to be associated with egg production traits. These results suggest that the identified SNPs in these genes might be useful in a selective breeding program to enhance productivity in warm environments.

## 1. Introduction

In Bangladesh, indigenous poultry farming plays a vital role in providing nutrition to much of the population, with almost 89% of rural households engaged in this sector [1]. Commercial strains and indigenous chicken breeds contribute almost equal numbers of eggs (50:50) and meat (60:40) to satisfy the national demand in Bangladesh [2,3]. Compared to the other indigenous breeds, the hilly chicken exhibits superior disease resistance, earlier sexual maturity, and higher egg production in Bangladesh’s hot weather. In addition, the hilly chicken shows lower mortality during rearing in rural areas, highlighting its potential as a promising indigenous chicken breed in Bangladesh [4].

However, due to global climate change, indigenous poultry farming in Bangladesh faces the economically harmful challenge of high ambient temperatures [5]. During heat stress, birds generally thermoregulate through reduced feed intake, hormonal regulation, and panting, which can negatively influence production and reproduction [6,7]. Chicken thermoregulation also involves the expression of heat shock-related genes, such as genes that encode heat shock proteins (HSPs) and heat shock transcription factors (HSF) [8,9].

The heat shock protein 70 (HSP70) gene, a member of the HSP family, is expressed in almost all types of cells [10]. This gene plays protective roles in various stress responses, including heat stress, and maintains homeostasis by balancing the synthesis and degradation of cellular proteins [11,12]. It has been demonstrated in previous studies that genetic variations (i.e., single-nucleotide polymorphisms; SNPs) in the 5′-flanking regions of *HSP70* affect the function of this gene, leading to changes in cellular processes and influencing phenotypic traits, including heat tolerance, weaning weight, milk production, fertility, and disease susceptibility in large animals [13,14]. Similarly, certain SNPs (e.g., A258G, C276G and C1431A) in the coding region of *HSP70* have been linked to thermotolerance, production, and reproductive performance in chickens [15,16]. Polymorphisms in the 5′-flanking regions of *HSP70* have also been associated with thermotolerance and reproductive traits in mammals [17,18,19]; however, the correlation between these SNPs and egg production traits in chickens remains unclear.

Heat shock proteins, including HSP70, are transcriptionally regulated by HSF3, a member of the HSF protein family, in chickens [20,21]. *HSF3* regulates the expression of HSP70 and acts as a primary defense against heat stress [21,22]. A previous study found that the genetic polymorphism A-1388G alters the activity of the CdxA transcription binding site, resulting in changes to the promoter activity of the *HSF3* gene in chickens [23,24]. This alteration has also been associated with heat-resistance parameters in Lingshan chickens; however, the associations between chicken reproductive traits and SNPs of *HSF3* remain unexplored [23].

With regard to indigenous breeds such as the hilly chicken of Bangladesh, which exhibit resilience to harsh conditions, the genetic mechanisms underlying their superior thermotolerance, particularly in relation to HSPs and HSFs, remain inadequately explored [25]. In particular, understanding how genetic variations in key genes such as *HSP70* and *HSF3* affect reproductive traits under heat stress is crucial to optimizing poultry breeding programs for improved productivity in tropical climates [26,27].

We hypothesized that genetic variations in the *HSP70* and *HSF3* genes contribute to egg production traits in hilly chickens and could be used to improve thermotolerance and productivity in chickens raised in high ambient temperatures. Therefore, we aimed (i) to identify genetic variations (SNPs) in the *HSP70* and *HSF3* genes in hilly chicken in Bangladesh and (ii) to assess the associations between these SNPs and egg production traits. The development of genetic selection strategies aimed at improving thermotolerance and reproductive performance in poultry could help to improve the sustainability of chicken farming in hot climates.

## 2. Materials and Methods

### 2.1. Experimental Birds and Trait Records

A total of 150 female hilly chickens maintained at the Bangladesh Livestock Research Institute (BLRI) (9th generation of the breeding flock) were used in the present study (Figure 1). These hens were reared from hatching under the standard management protocol of the BLRI. At 16 weeks of age, the birds were transferred to separate cages in a naturally ventilated poultry house with a 16 h photoperiod that included 12 h of daylight and 4 h of artificial light. During the laying age period (17–72 weeks), the birds were fed twice a day (morning and evening) with a diet containing 16.33% crude protein and 2845 kcal ME/kg DM. The hens were provided free access to water. From that point until reaching 310 days of age, the following parameters were recorded: the age at sexual maturity (ASM), body weight (BW) at ASM, egg weight (EW) at ASM (g), monthly egg production (number/bird), and EW at 40 weeks of age (g).

### 2.2. Blood Collection and Genomic DNA Extraction

Blood samples from mature hilly hens were collected at 310 days of age and stored on FTA cards (Qiagen GmbH, Hilden, Germany). The genomic DNA was extracted from the cards according to the manufacturer’s instructions. After the genomic DNA was extracted, the DNA concentrations were measured using a Bio Spec-Nano (Shimadzu Corp., Kyoto, Japan) and stored at 20 °C until further analysis.

### 2.3. Primer Design, PCR Amplification, and Sequencing of the HSP70 Fragment

A pair of primers (Table 1) was designed using Primer3 software, v4.1.0(NCBI), utilizing the complete DNA sequence of HSP70 (NC_052536.1). To amplify the 5′ flanking region of the HSP70 gene, PCR was performed in a final reaction volume of 20 μL, containing 100 ng of genomic DNA, 10 μL of 2× GoTaq Green Master Mix (Promega Corp., Madison, WI, USA), and 10 pmol/μL of each primer (HSP70_F_Common and HSP70_R). Amplification was conducted, beginning with initial denaturation at 94 °C for 5 min, followed by 30 cycles of denaturation at 94 °C for 30 s, annealing at 64 °C for 30 s, and extension at 72 °C for 30 s, then a final 5 min of extension at 72 °C. The PCR products were electrophoresed on a 1% agarose gel, and the gel was stained with ethidium bromide to visualize the amplicons. The amplified PCR products were purified from the agarose gel using a NucleoSpin Gel and PCR Clean-up Kit (Macherey–Nagel GmbH & Co. KG, Valencienner, Dueren, Germany). The purified DNA provided the template for direct sequencing using a Big Dye Terminator v3.1 Cycle Sequencing Kit (Applied Biosystems, Inc., Foster City, CA, USA) for each primer strand (HSP70_F/Seq and HSP70_R/Seq) (Table 1). An AB1 3130 Sequencer (Applied Biosystems) was used to sequence the products according to the manufacturer’s protocol. The sequence data for the 5′-flanking regions of HSP70 were edited, assembled, and aligned. Polymorphism detection was conducted using GENETYX v.15 (GENETYX Corp., Tokyo, Japan).

### 2.4. Genotyping of SNPs and Reconstruction of Haplotypes in the HSP70 Gene

The SNPs were genotyped separately using allele-specific PCR (AS-PCR). AS-PCR was performed in a 20 μL reaction volume containing 10 pmol/μL of each primer (Table 1) using combinations of P1 + P5 or P2 + P5 for G-399A, P3 + P5 or P4 + P5 for A 68G, P9 + P13 or P10 + P13 for A258G, P11 + P13 or P12 + P13 for C276G, and P14 + P16 or P15 + P16 for C1431A. For the PCR reaction, we used 100 ng of DNA and 10 μL of 2 × GoTaq Green Master Mix (Promega); the remainder of the reaction volume was made up using nuclease-free water. The PCR protocol consisted of the following steps: initial denaturation at 94 °C for 5 min, denaturation at 94 °C for 30 s, annealing temperatures and cycle numbers as in Table 1, extension at 72 °C for 30 s, and a final extension at 72 °C for 5 min. The PCR products were separated on 1.5% (>200 bp) or 2% (<200 bp) agarose gel depending on the amplicon size, and the gel was stained with ethidium bromide for visualization. Haplotypes were constructed using the population genotyping data (Table 2).

### 2.5. Genotyping of SNPs and Reconstruction of Haplotypes Within HSF3 Gene

Genotyping of the SNPs within HSF3 was performed via AS-PCR. The primer (Table 3) combinations of P17 + P19 or P18 + P19 were used for A-1388G, and those of P20 + P22 or P21 + P22 were used for A-1703G. The PCR conditions and mixtures are detailed above (Section 2.4). Haplotypes were constructed using genotyping data as described in Table 2.

### 2.6. Statistical Analysis

To assess the associations between egg production traits and SNPs or haplotypes, a general linear model procedure in IBM SPSS Statistics for Windows, v.20.0 (IBM Corp., Armonk, NY, USA) was used. The following equation was employed for the analysis [28]:Y_ij_ = μ + G_i_+ e_ij_
where Y_ij_ represents the phenotypic value of the specific traits (e.g., egg production, EW, ASM, and BW), μ represents the population mean of the target trait, G_i_ represents the genotype effect (where i = 3 genotypes), and e_ij_ represents the random residual error associated with the Y_ij_ observation. A chi-squared (χ^2^) test was used to assess the Hardy–Weinberg equilibrium (HWE) in the population. The parameter values were presented as the least square means ± standard error, and statistical significance (least significant difference) was evaluated at *p* < 0.05. Haplotype frequencies with a minimum threshold of >3% were considered for the association study.

## 3. Results

### 3.1. Identification of Novel SNPs in the 5′-Flanking Region of HSP70

Based on a reference sequence of HSP70 from the NCBI database (https://www.ncbi.nlm.nih.gov/, accessed on 2 July 2022; NC_052536.1), two novel SNPs, G-399A (Figure 2A) and A-68G (Figure 2B), were identified in the 729 bp length of the 5′-flanking regions of the HSP70 gene in Bangladeshi hilly chickens.

### 3.2. Genotypic and Allelic Frequencies and Haplotype Combinations in HSP70

A total of five SNPs (Table 4) were genotyped, including two novel SNPs, G 399A (Figure 3A) and A-68G (Figure 3B), and three previously reported SNPs, A258G (Figure 3C), C276G (Figure 3D), and C1431A (Figure 3E), in the HSP70 gene of hilly chickens.

Table 5 shows the genotypic and allelic frequencies for the analyzed SNPs in HSP70. For the G-399A SNP, the frequency of the wild-type GG genotype (0.92) was significantly (*p* < 0.05) greater than that of the AG (0.08) genotype, and the frequency of the G allele (0.95) was notably greater (*p* < 0.05) than that of its A allele (0.05) counterpart. Regarding the A-68G SNP, the frequency of the AA genotype (0.67) was significantly greater (*p* < 0.05) compared with the AG (0.22) and GG (0.10) genotypes, and the frequency of the A allele (0.78) was greater than that of the G allele (0.22).

In SNP A258G, the frequency of the AG genotype (0.77) was significantly higher (*p* < 0.05) than that of the AA genotype (0.20) or the GG genotype (0.03). The frequency of the A allele (0.58) was comparatively greater (*p* < 0.05) than that of the G allele (0.4) for this SNP. Regarding the C276G SNP, the CC genotype had a more significant (*p* < 0.05) frequency (0.58) than the CG (0.39) and GG (0.03) genotypes. In addition, an increased frequency of the C allele (0.77) was noted compared with the G allele (0.23). The C1431A SNP of *HSP70* exhibited two genotypes: CC (0.74) and CA (0.26). The frequency of the C allele (0.87) was significantly higher (*p* < 0.05) than that of the A allele (0.13). All SNPs, except C276G and C1431A, were outside the HWE (*p* < 0.05). Table 2 presents the 15 distinct haplotypes (H1–H15) identified in the study sample, derived from the genotyping data of the five SNPs. The most common haplotype was H1, with a frequency of 0.34, while the frequencies of the other haplotypes ranged from 0.02 to 0.12.

### 3.3. Association Between Genotypes and Egg Production Traits for HSP70

Table 6 shows the associations between specific polymorphisms in the *HSP70* gene and egg production traits in chickens. Significant effects were observed for certain SNPs on traits such as egg number (EN) and EW. Notably, G-399A and A258G were linked to egg production during specific intervals, while A-68G was associated with EW at 40 weeks (Table 6).

G-399A was significantly (*p* < 0.05) correlated with EN at 161–190 days of age. Birds with the AG genotype of this SNP had significantly (*p* < 0.05) greater ENs compared to birds with the GG genotype. The A-68G SNP demonstrated a significant association with egg weight (EW) at 40 weeks of age, with birds possessing the mutant GG genotype exhibiting a notably higher EW (*p* < 0.05) in comparison to those with the wild AA genotype (Table 7).

The A258G SNP was related (*p* < 0.05) to the EN at 221–250 days of age. In addition, the AA genotype hens produced the highest number of eggs (16.00), which was significantly greater than the other two genotypes; however, no significant variations were observed between the AG and GG genotypes. The C276G SNP did not exhibit a significant association with the EN at 281–310 days of age; nevertheless, chickens with the GG genotype had statistically higher ENs compared to chickens with the CC and CG genotypes. The C1431A SNP was found to be associated (*p* < 0.05) with EN at 191–220 days of age, and the birds with the heterozygous CA genotype had significantly higher ENs than the CC genotype birds (Table 7).

### 3.4. Association of the Haplotypes for HSP70 with Egg Production Traits in the Hilly Chicken

This study considered a total of eight haplotypes (H1–H3 and H7–H11) with frequencies of >3% that were used in the association analysis. These haplotypes were significantly (*p* < 0.05) associated with ASM, EW at ASM, EW at 40 weeks, BW at ASM, and EN. Among all haplotypes, H11 exhibited an association with a substantially earlier ASM and a greater EN at 130–160 and 161–190 days of age compared to the other haplotypes. Haplotype 2 (H2) was significantly associated with reduced BW at ASM and increased EN at 191–220 days of age relative to the other haplotypes. In addition, H7 exhibited an association with markedly elevated EW values at ASM and 40 weeks compared with the other haplotypes (Table 8).

### 3.5. Genotypic and Allelic Frequencies and Haplotype Combinations in HSF3

Two previously known SNPs, A-1388G (Figure 4A) and A-1703G (Figure 4B), in the 5′-untranslated region (UTR) of the *HSF3* gene were genotyped in the studied flock.

Table 9 presents the genotype and allele frequencies and reveals that the A-1388G SNP showed only two categories of genotypes, with the frequency of the dominant AA genotype (0.94) being remarkably greater than that of the AG genotype (0.06). The frequency of allele A (0.97) was much higher than that of the G allele (0.03), and the allocation of genotypes in the studied flock did not conflict with the HWE (*p* > 0.05).

Regarding the A-1703G SNP, the frequency of the AA genotype was notably greater (0.91) than that of the AG genotype (0.09), with a significantly greater frequency of the A allele (0.96) compared with the G allele (0.04). In addition, according to the χ^2^ test, this SNP deviated from the HWE (*p* < 0.05). The genotype data were used to perform haplotype reconstruction, which revealed the existence of two haplotype categories, AA and AG, from two novel SNPs (G-399A and A-68G), among the 150 individual birds that were examined.

### 3.6. Association Between Genotypes and Egg Production Traits and HSF3

The significant association analyses between the SNPs and egg production traits are shown in Table 10. For SNP A-1388G, a significant (*p* < 0.05) association was found for the EN at 130–160 days of age, with a greater value observed for the AG genotype compared with the AA genotype.

However, a significantly reduced EN was also found at 251–280 days of age for this SNP (A-1388G) compared with the AA genotypes (Table 11). Lastly, the A-1703G SNP did not show any significant (*p* < 0.05) correlations with the egg production traits in the studied flock (Table 10).

### 3.7. Association of HSF3 Haplotypes with Hilly Chicken Egg Production Traits

A significant association (*p* < 0.05) was observed between the haplotypes and the ASM and ENs at different ages. Compared with H1, H2 was associated with a significantly earlier ASM (days) and a higher EN during the 130–160 days of age period. Meanwhile, H1 showed a significant correlation with a higher EN during the 251–280 days of age period compared with H2 (Table 12).

### 3.8. Evaluation of Combined Genotypic Effects of SNPs G-399A and A-68G in HSP70 with A-1388G SNP in HSF3 on Egg Production Traits

We analyzed the effects of the combined genotypes of G-399A/A-1388G and A-68G/A-1388G SNPs on the phenotypic performance of the studied population. Birds with wild-type/heterozygote combinations for the G-399A and A-1388G SNPs showed a significantly (*p* < 0.05) earlier ASM and higher ENs at 130–160 days of age compared to wild-type/wild-type birds. BW at 40 weeks and EN at 161–190 and 251–280 days of age were also significantly (*p* < 0.05) influenced by different combinations of these two SNPs (Table 13).

The combined genotypic effects of the A-68G and A-1388G SNPs indicate that any mutant combinations, such as Het × Wild (AG × AA) or Mut × Wild (GG × AA), exhibited a more pronounced impact on EW at 40 weeks and ASM compared to Wild × Wild (AA × AA) birds (*p* < 0.05) (Table 14).

## 4. Discussion

High ambient temperatures negatively affect livestock reproduction, and this impact may be intensified by ongoing global warming, particularly for backyard poultry farms. The use of genetic heat-resistance markers in animal breeding programs is beneficial for improving poultry productivity in hot climates [30,31].

HSP70 is one of the most common biological response markers of thermal stress and participates in numerous physiological processes, including protein folding, transportation, and assembly within cells [32,33]. It also protects cells by inhibiting the apoptotic pathway, which can positively impact animal health and productivity [34]. The present study identified SNPs in the *HSP70* and *HSFF3* genes that regulate the transcription of *HSP70* and investigated their associations with the reproductive traits of Bangladeshi hilly chickens.

Analysis of the *HSP70* gene in hilly chickens showed that it exhibited heterogeneity in the 5′-flanking regions. These genetic variations are caused by the transitions of the nucleotides at positions −399 bp and −68 bp 5′-upstream from the start codon. The 5′-flanking region of *HSP70* is polymorphic, and several SNPs have also been reported in the naked neck broiler and in Indonesian local chickens [18,35,36]. In addition, three previously reported synonymous SNPs in the coding region of this gene (A258G, C276G and C1431A) were also found in the studied flock. All the SNPs in *HSP70*, except C276G and C1431A, were outside the Hardy–Weinberg equilibrium (Table 3). This might be due to the selection and breeding strategy used to improve the studied flock.

The present study revealed significant associations between the tested *HSP70* SNPs and specific egg production traits in hilly chickens (Table 6). The novel G-399A SNP was found to be significantly correlated with greater ENs (*p* < 0.05), and A-68G showed a significant association with increased EW (*p* < 0.05) (Table 7). These associations indicated that genetic variations in *HSP70* may influence egg production efficiency in chickens in hot environments. Similar effects associated with *HSP70* SNPs in the 5′-flanking region have been reported in other vertebrates, including the finding that several SNPs in the 5′-flanking region of the *HSP70* gene in cattle were associated with mRNA stability, stress responses, milk production, and calf weaning weight [14,17]. Variations in the 5′-flanking regions of *HSP70* may alter the specific binding sites of the transcription factors and could therefore modulate the binding efficiency that affects gene functions. This could lead to changes in cellular processes and ultimately alter phenotype performance [14,37]. The precise mechanism of how *HSP70* affects egg production in chickens remains unknown. However, a possible explanation might be that during heat stress, HSP70 regulates thermotolerance and inhibits apoptosis in ovarian cells, which may lead to the protection of granulosa cells, ultimately improving folliculogenesis and egg production [38,39,40]. Therefore, further physiological research is required in this area.

Regarding the three examined synonymous SNPs of *HSP70*, the AA genotype of A258G conferred a higher EN (*p* < 0.05) at 221–250 days than the other genotypes. The AA genotype of the same SNP was significantly associated with improved thermotolerance and BW at an early stage in Taiwanese chickens, but not with EW or EN until 280 days of age [16]. This discrepancy may be due to differences in chicken breeds or environmental factors such as temperature and duration of heat stress exposure. Furthermore, the GG genotype of the C276G SNP was significantly associated with greater EN (Table 7). In a previous study, the C276G polymorphism was found to produce a novel haplotype in combination with the A258G SNP, reported as a heat stress marker in Indonesian Walik chickens [15]. These silent mutations in the coding region of the *HSP70* gene have been previously reported as heat-resistance markers in chickens [41]. However, this earlier study was not an association study considering reproductive performance in chickens. In the present study, the C1431A SNP was significantly correlated with increased ENs, as shown in Table 7. A significant association between this SNP and EW, fertility, and hatchability percentage was previously observed in Iranian Mazandaran native breeder chickens, but the authors did not report an association between this SNP and the EN [29].

Although important egg production traits such as ASM, EW at ASM, and BW at ASM were not correlated with any individual *HSP70* SNPs (Table 6), the haplotype combinations resulting from the five SNPs significantly affected these traits (Table 8). This suggests that the individual effects of these SNPs are small compared with their combined effects. In chickens, combined genotypes have a greater impact on BW than individual genotypes [42].

Regarding the SNPs in the *HSF3* gene, although the A-1703G SNP did not significantly affect the egg production traits in the studied flock, the A-1388G SNP was found to be significantly associated with the EN (Table 11). Notably, the A-1388G polymorphism has been found to change the CdxA transcription factor binding site associated with thermotolerance in Chinese Lingshan chickens [23]. Haplotype analysis of *HSF3* also revealed significant associations with several egg production traits, including EN and ASM for H2 (Table 12). Laying hens with the H2 haplotype exhibited an earlier ASM and significantly greater ENs than hens with the H1 haplotype. However, it is noteworthy that the EN was reduced at a later stage (251–280 days of age) for H2, indicating that this haplotype may be less advantageous for long-term egg production.

Our findings revealed significant associations between specific SNPs within the *HSP70* and *HSF3* genes and egg production traits in the studied population. Analysis of the combined effects of two novel SNPs (G-399A and A-68G) in *HSP70* and the A-1388G SNP in *HSF3* on phenotypic performance revealed a significant interaction that suggests that these genes may play synergistic or compensatory roles in egg production in hot environments.

## 5. Conclusions

Two novel SNPs (G-399A and A-68G) were identified in the 5′-flanking regions of *HSP70* in hilly chickens, and three previously known SNPs (A258G, C276G and C1431A) of *HSP70* also existed in *HSP70* in the same breed. In addition, the *HSF3* gene was also found to possess two reported SNPs (A-1388G and A-1703G) in this studied breed. All the SNPs in both genes, except A-1703G in *HSF3*, and their corresponding haplotypes were associated with egg production traits in hilly chickens. These significant SNPs and haplotypes might be available in the future for molecular marker-based selection programs to enhance the egg productivity of hilly chickens in the high ambient temperatures of Bangladesh. Further research is needed to elucidate the precise mechanisms that enable HSPs to improve the productive performance of chickens.

## Figures and Tables

**Figure 1 animals-14-03552-f001:**
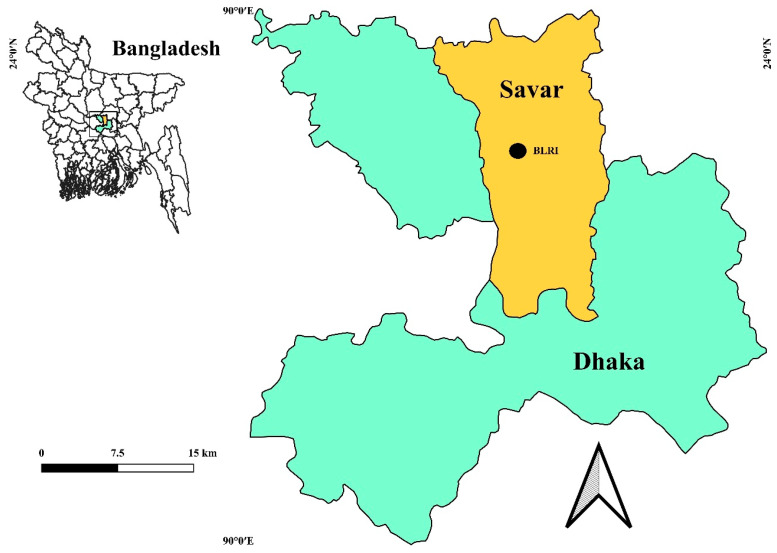
Location of the conducted study.

**Figure 2 animals-14-03552-f002:**
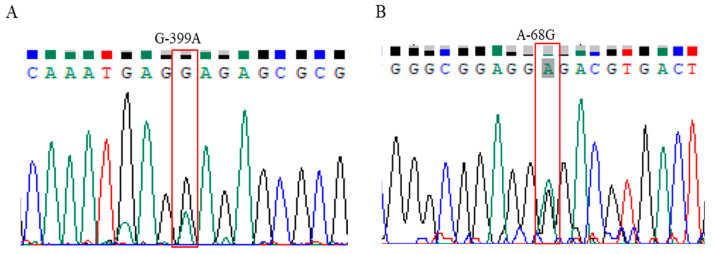
Determination of SNPs in the 5′-flanking regions of the *HSP70* gene by sequencing. (**A**) The unique SNP G-399A was detected as multiple peaks at the same positions of 399 bp upstream from the start codon. (**B**) The unique SNP A-68G was detected as multiple peaks at the same positions of 68 bp upstream from the start codon.

**Figure 3 animals-14-03552-f003:**
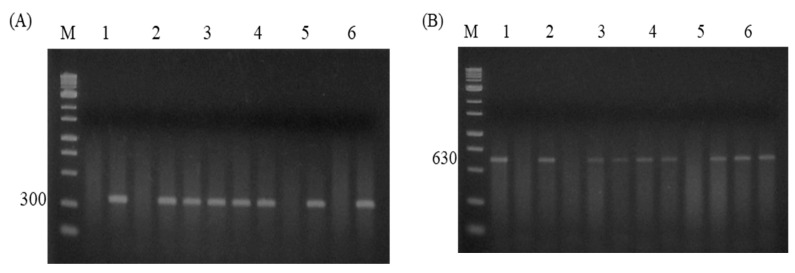
The electrophoresis images of AS-PCR for five SNPs, including two novel SNPs (G-399A and A-68G) and three known in the coding regions (CDs) of the *HSP70* gene. (**A**) Image of AS-PCR for G-399A SNP generated 300 bp. Samples 1, 2, and 5 represent wild GG, while 3 and 4 represent AG genotypes, respectively. M shows a 1 kb DNA ladder marker. (**B**) Photograph of AS-PCR for the A-68G SNP, which produced 630 bp. AA (Samples 1, 2), AG (Samples 3, 4 and 6) and GG (Sample 5) represent mutated homozygous genotypes. M shows a 1 kb DNA ladder marker. (**C**) Photograph of AS-PCR for the A258G SNP, which produced 600 bp. AA (Sample 1) represented wild type, while AG (Sample 2) indicates heterozygous, and mutated homozygous GG is represented in Sample 3, respectively. M shows a 1 kb DNA ladder marker. (**D**) The image of AS-PCR for the C276G SNP yielded 616 bp. Mutated GG (Sample 4) genotypes, CC (Samples 1 and 3), and CG (Sample 2) indicate wild and heterozygous genotypes, respectively. M shows a 1 kb DNA ladder marker. (**E**) Picture of AS-PCR for the SNP of C1431A, which created 198 bp. Heterozygous CA (Samples 1–3) genotype, while CC (Sample 4) represents a wild genotype. M shows a 100 bp DNA ladder marker.

**Figure 4 animals-14-03552-f004:**
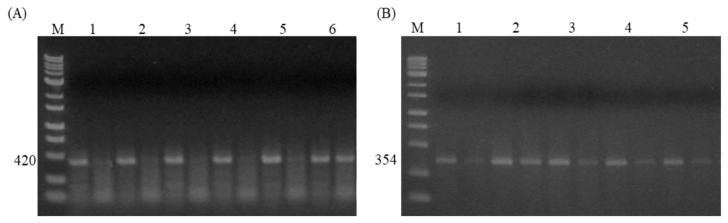
Electrophoresis images of AS-PCR for two SNPs in the 5′-UTR regions of *HSF3* gene. (**A**) Photograph of AS-PCR for the A-1388G SNP, which produced 420 bp. Only two categorized genotypes, AA (Samples 1–5) and AG (Sample 6), were found. M shows a 1 kb DNA ladder marker. (**B**) Image of AS-PCR for the SNP A-1703G, which produced 354 bp. Samples 1, 3, 4, and 5 represent AA, while sample 2 indicates AG genotype. M shows a 1 kb DNA ladder marker.

**Table 1 animals-14-03552-t001:** Primer sequences for sequencing and genotyping of the *HSP70* gene.

No	Primer Name	Sequence (5′ to 3′)	Anneal. Temp.	PCR Cycle	Product Length
P1 P2 P3 P4 P5 P6 P7 P8	*HSP70*_R_S1_G *HSP70*_R_S1_A *HSP70*_R_S2_A *HSP70*_R_S2_G *HSP70*_F_Com *HSP70*_R *HSP70*_F_Seq *HSP70*_R_Seq	CCAATCACAACGCGCTCTC CCAATCACAACGCGCTCTT TCGCTCGCAGTCACGTCT TCGCTCGCAGTCACGTCC AGAAGTTGTGTGAGTCGCGA AATACGTGGTGCCCAGATCG GTCGCGACCAAATAAGGGTA GTGCCCAGATCGATGCCGATG		30	300

			64	30	630

				30	729

P9 P10 P11 P12 P13	*HSP70*_R_S1_A *HSP70*_R_S1_G *HSP70*_R_S2_C *HSP70*_R_S2_G *HSP70*_F_Com	GAAGGGCCAGTGCTTCATGTCT GAAGGGCCAGTGCTTCATGTCC CCTCGTTCACCACACGGAAG CCTCGTTCACCACACGGAAC CGATCTGGCTGCAATCTACG	60	30	600

			58	30	616

P14 P15 P16	*HSP70*_R_S3_C *HSP70*_R_S3_A *HSP70*_S3_F_Com	CTATGTCAAAAGTGACCTCG CTATGTCAAAAGTGACCTCT AGCGTAACACCACCATTC	55	30	198


F: Forward primer, R: Reverse primer, Com: Common. Primer sets P1–P6 and P9–P16 were used for the amplification of specific alleles, and primers P7–P8 were used for sequencing the 5′ flanking in *HSP70*.

**Table 2 animals-14-03552-t002:** Haplotype construction using 5 SNPs in the *HSP70* gene and their frequencies.

Haplotype	Position of SNP					Frequency
	G-399A	A-68G	A258G	C276G	C1431A	
H1	G	A	A	C	C	0.34
H2	G	A	A	C	A	0.04
H3	G	A	G	G	C	0.08
H4	G	A	G	C	A	0.03
H5	A	A	A	C	C	0.02
H6	G	A	A	G	C	0.03
H7	G	G	A	C	C	0.09
H8	G	A	G	G	A	0.07
H9	G	A	G	C	C	0.12
H10	G	G	G	C	C	0.03
H11	G	G	G	C	A	0.04
H12	G	G	G	G	A	0.02
H13	G	G	A	G	A	0.03
H14	G	G	A	C	A	0.03
H15	G	G	G	G	C	0.03

**Table 3 animals-14-03552-t003:** Primer sequences for genotyping of *HSF3* gene.

No	Primer Name	Sequence (5′ to 3′)	Anneal. Temp.	PCR Cycle	Product Length
P17 P18 P19	*HSF3*_F_S1_A *HSF3*_F_S1_G *HSF3*_S1_R_Com	GTCCCCATAATACCTCCCCA GTCCCCATAATACCTCCCCG TTTTAGCTGCCAGTTCCTTT	60	25	354
P20 P21 P22	*HSF3*_R_S2_ A *HSF3*_R_S2_G *HSF3*_S2_F_Com	TTTTAGCTGCCAGTTCCTTT TTTTAGCTGCCAGTTCCTTC AAGAATGGCTCCTTGCCACC	59	30	420

F: Forward primer, R: Reverse primer, Com: Common. Primer sets P17–P22 were used for the amplification of specific alleles in the *HSF3* gene.

**Table 4 animals-14-03552-t004:** Analyzed single-nucleotide polymorphisms (SNPs) in the chicken HSP70 gene.

SNP Name	Mutation	Location	Genomic Position
G-399A	G>A *	5′-flanking	52383334
A-68G	A>G *	5′-flanking	52383665
A258G	A>G	Coding	[15]
C276G	C>G	Coding	[15]
C1431A	C>A	Coding	[29]

* Novel SNP found in the HSP70 gene.

**Table 5 animals-14-03552-t005:** Genotypic and allelic frequencies with Hardy-Weinberg equilibrium test at the SNP locus of the *HSP70* gene.

SNPs	Genotype Frequency			Allele Frequency		χ^2 (HWE)^	*p*-Value
G-399A	GG	AG	AA	G	A		
	0.92(137)	0.08(12)	–	0.95	0.05	7.44	*p* < 0.01
A-68G	AA	AG	GG	A	G		
	0.67(72)	0.22(24)	0.10(11)	0.78	0.22	11.68	*p* < 0.01
A258G	AA	AG	GG	A	G		
	0.20(28)	0.77(113)	0.03(5)	0.58	0.42	50.44	*p* < 0.00
C276G	CC	CG	GG	C	G		
	0.58(86)	0.39(58)	0.03(5)	0.77	0.23	1.63	*p* > 0.05
C1431A	CC	CA	AA	C	A		
	0.74(110)	0.26(39)	–	0.87	0.13	3.42	*p* > 0.05

SNP: Single nucleotide polymorphism, *p* < 0.05: statistically significant using Pearson’s χ^2^ test, *p* > 0.05: Non-significant, HWE: Hardy–Weinberg equilibrium.

**Table 6 animals-14-03552-t006:** Association of polymorphisms in *HSP70* gene with egg production traits.

SNPs	Traits (*p* Value of Significant Test)									
	ASM (Days)	BW at ASM	EW at ASM	EW at 40 WK	EN 130–160 d	EN 161–190 d	EN 191–220 d	EN 221–250 d	EN 251–280 d	EN 281–310 d
G-399A	0.890	0.148	0.129	0.660	0.578	0.020	0.421	0.218	0.484	0.225
A-68G	0.365	0.537	0.707	0.009	0.547	0.520	0.478	0.444	0.413	0.186
A258G	0.543	0.593	0.426	0.960	0.561	0.563	0.590	0.016	0.377	0.805
C276G	0.382	0.246	0.275	0.795	0.568	0.304	0.759	0.262	0.222	0.050
C1431A	0.121	0.464	0.651	0.855	0.969	0.681	0.012	0.058	0.341	0.363

Significant: *p* < 0.05, ASM: Age at sexual maturity, EW: Egg weight, EN: Egg number, WK: Week.

**Table 7 animals-14-03552-t007:** Genotypic effects of SNPs in *HSP70* gene on egg production traits.

SNPs	Traits	Genotypes (Mean ± SE)			*p* Value
G-399A	EN at 161–190 d	GG 16.27 ± 0.26 ^b^	AG 18.83 ± 0.86 ^a^	AA –	0.020
A-68G	EW at 40 wk	AA 46.56 ± 0.11 ^b^	AG 47.03 ± 0.18 ^ab^	GG 47.30 ± 0.27 ^a^	0.009
A258G	EN at 221–250 d	AA 16.00 ± 0.19 ^a^	AG 14.82 ± 0.63 ^b^	GG 14.60 ± 1.53 ^b^	0.016
C276G	EN at 281–310 d	CC 13.01 ± 0.24	CG 12.67 ± 0.29	GG 13.60 ± 0.93	0.050
C1431A	EN at 191–220 d	CC 16.49 ± 0.21 ^b^	CA 17.61 ± 0.39 ^a^	AA –	0.012

^ab^ Means with different superscripts within the same row differ significantly. *p* < 0.05: statistically significant, EW: Egg weight, EN: Egg number.

**Table 8 animals-14-03552-t008:** Association of haplotypes of the *HSP70* polymorphisms with egg production traits in Hilly chicken.

Haplotypes	Traits (Mean ± SE)						
	ASM	EW at ASM	EW at 40 wk	BW at ASM	EN 130–160 d	EN 161–190 d	EN 191–220 d
H1(GAACC)	160.71 ± 1.61 ^ab^	25.85 ± 0.52 ^b^	46.812 ± 0.16 ^ab^	1728.52 ± 24.49 ^b^	2.00 ± 0.74 ^b^	15.91 ± 0.54 ^b^	16.03 ± 0.39 ^b^
H2(GAACA)	156.75 ± 4.68 ^ab^	26.00 ± 0.74 ^ab^	45.915 ± 0.46 ^b^	1543.75 ± 77.24 ^a^	4.50 ± 2.15 ^ab^	17.25 ± 1.58 ^ab^	18.75 ± 1.13 ^a^
H3(GAGGC)	157.00 ± 5.41 ^ab^	26.62 ± 0.52 ^ab^	46.356 ± 0.33 ^b^	1648.25 ± 54.62 ^ab^	4.25 ± 1.53 ^ab^	16.25 ± 1.12 ^ab^	16.75 ± 0.80 ^ab^
H7(GGACC)	161.44 ± 3.12 ^ab^	27.00 ± 0.49 ^a^	47.342 ± 0.31 ^a^	1839.55 ± 51.49 ^c^	4.33 ± 1.44 ^ab^	14.66 ± 1.05 ^ab^	16.44 ± 0.75 ^ab^
H8(GAGGA)	165.28 ± 3.54 ^b^	25.85 ± 0.55 ^ab^	46.526 ± 0.35 ^ab^	1688.85 ± 58.39 ^abc^	1.71 ± 1.63 ^ab^	15.57 ± 1.19 ^ab^	17.57 ± 0.86 ^ab^
H9(GAGCC)	158.88 ± 3.12 ^ab^	26.33 ± 0.49 ^ab^	46.399 ± 0.31 ^b^	1778.88 ± 51.49 ^c^	2.66 ± 1.43 ^ab^	17.22 ± 1.05 ^ab^	17.11 ± 0.75 ^ab^
H10(GGGCC)	159.25 ± 4.68 ^ab^	26.75 ± 0.74 ^ab^	47.125 ± 0.46 ^ab^	1711.25 ± 77.24 ^abc^	2.75 ± 2.15 ^ab^	16.75 ± 1.58 ^ab^	16.75 ± 1.13 ^ab^
H11(GGGCA)	153.25 ± 4.68 ^a^	27.00 ± 0.73 ^ab^	46.853 ± 0.46 ^ab^	1685.25 ± 77.24 ^abc^	7.00 ± 2.16 ^a^	18.25 ± 1.58 ^a^	18.25 ± 1.13 ^ab^

^abc^ Means with different superscripts within a column differ significantly (*p* < 0.05). ASM: Age at sexual maturity, EW: Egg weight (g), BW: Body weight (g), EN: Egg number/bird/month, WK: Week.

**Table 9 animals-14-03552-t009:** Genotypic and allelic frequencies with Hardy–Weinberg equilibrium (HWE) test at the SNP locus of *HSF3* gene.

SNPs	Genotype Frequency			Allele Frequency		χ^2 (HWE)^	*p*-Value
A-1388G	AA	AG	GG	A	G		
	0.94(141)	0.06(9)	–	0.97	0.03	0.143	*p* > 0.05
A-1703G	AA	AG	GG	A	G		
	0.91(136)	0.09(14)	–	0.96	0.04	6.88	*p* < 0.01

SNP: Single nucleotide polymorphism, *p* < 0.05: statistically significant using Pearson’s χ^2^ test, *p* > 0.05: Non-significant, *HWE*: Hardy–Weinberg equilibrium.

**Table 10 animals-14-03552-t010:** Association of polymorphisms in *HSF3* gene with egg production traits.

SNPs	Traits (*p* Value of Significant Test)									
	ASM (Days)	BW at ASM(g)	EW at ASM(g)	EW at 40 wk(g)	EN 130–160 d	EN 161–190 d	EN 191–220 d	EN 221–250 d	EN 251–280 d	EN 281–310 d
A-1388G	0.071	0.948	0.679	0.450	0.037	0.803	0.083	0.913	0.013	0.200
A-1703G	0.363	0.484	0.406	0.510	0.731	0.572	0.342	0.569	0.598	0.818

Significant: *p* < 0.05, ASM: Age at sexual maturity, EW: Egg weight, EN: Egg number, WK: Week.

**Table 11 animals-14-03552-t011:** Effects of SNPs in *HSF3* gene on egg production.

SNPs	Traits	Genotypes (Mean ± SE)			*p* Value
A-1388G		AA	AG	–	
	EN at 130–160 d EN at 251–280 d	2.85 ± 0.36 ^a^ 11.04 ± 0.25 ^a^	5.55 ± 1.22 ^b^ 8.82 ± 0.84 ^b^	– –	0.037 0.013

^ab^ Means with different superscripts within the same row differ significantly (*p* < 0.05), EN: Egg number.

**Table 12 animals-14-03552-t012:** Association of haplotypes in *HSF3* gene with egg production traits in Hilly chicken.

Haplotypes	Traits (Mean ± SE)						
	ASM	EW at ASM	EW at 40 WK	BW at ASM	EN 130–160 d	EN 161–190 d	EN 251–280 d
H1(AA)	159.95 ± 0.76 ^b^	26.14 ± 0.14	46.77 ± 0.08	1746.38 ± 14.54	2.76 ± 0.35 ^b^	16.48 ± 0.27	11.03 ± 0.25 ^a^
H2(AG)	155.44 ± 1.78 ^a^	25.96 ± 0.33	46.85 ± 0.18	1704.08 ± 34.18	4.60 ± 0.82 ^a^	16.61 ± 0.64	9.52 ± 0.59 ^b^

^ab^ Means within a column with different superscripts differ significantly (*p* < 0.05). ASM: Age at sexual maturity, EW: Egg weight (g), BW: Body weight (g), EN: Egg number/bird/month, WK: Week.

**Table 13 animals-14-03552-t013:** Combined genotypic effects of two SNPs (G-399A in HSP70 gene and A-1388G in HSF3 gene) on productive and reproductive performances.

Parameter	Genotype (Mean ± SE)				*p* Value
	Wild × Wild (GG × AA) 0.84 (120)	Wild × Het (GG × AG) 0.06 (9)	Het × Wild (AG × AA) 0.08 (11)	Mut × Wild (AA × AA) 0.01 (2)	
ASM (d)	160.08 ± 0.79 ^b^	154.11 ± 2.90 ^a^	156.0 ± 2.63 ^ab^	157.0 ± 6.16 ^ab^	0.05
EW at ASM (g)	26.16 ± 0.15	25.56 ± 0.53	25.91 ± 0.48	27.0 ± 1.13	0.28
BW at ASM (g)	1748.71 ± 14.11	1730.89 ± 51.51	1672.91 ± 46.60	1715.0 ± 109.27	0.74
BW at 40 Wks	2047.77 ± 24.65 ^a^	1947.78 ± 90.0 ^ab^	1838.73 ± 81.41 ^b^	2260.5 ± 190.92 ^a^	0.02
EW at 40 Wks	46.78 ± 0.08	46.39 ± 0.30	46.95 ± 0.28	46.72 ± 0.65	0.23
EN at 130–160 d	2.71 ± 0.36 ^b^	6.11 ± 1.33 ^a^	3.64 ± 1.20 ^ab^	3.49 ± 2.82 ^ab^	0.02
EN at 161–190 d	16.26 ± 0.28 ^b^	16.89 ± 1.02 ^ab^	18.91 ± 0.92 ^a^	16.50 ± 2.16 ^ab^	0.01
EN at 191–220 d	16.85 ± 0.20	15.44 ± 0.73	16.45 ± 0.66	18.5 ± 1.55	0.07
EN at 221–250 d	15.13 ± 0.19	15.0 ± 0.71	14.91 ± 0.64	12.50 ± 1.51	0.86
EN at 251–280 d	11.14 ± 0.25 ^a^	8.56 ± 0.92 ^b^	10.18 ± 0.84 ^ab^	12.5 ± 1.96 ^ab^	0.01
EN at 281–310 d	13.0 ± 0.19	13.67 ± 0.69	11.91 ± 0.63	13.0 ± 1.48	0.36

^ab^ Means with different superscripts within the same row differ significantly, Significant: *p* < 0.05, Non-significant: *p* > 0.05, Het: Heterozygous, Mut: Mutant, ASM: age at sexual maturity, EN: egg number/month, EW: egg weight, BW: body weight, Wks: weeks, HW: hatched weight.

**Table 14 animals-14-03552-t014:** Combined genotypic effects of A-68G SNP in HSP70 gene and A-1388G SNP in HSF3 gene on productive performance.

Parameter	Combined Genotypes (Mean ± SE)				*p* Value
	Wild × Wild (AA × AA) 0.62 (65)	Wild × Het (AA × AG) 0.07 (7)	Het × Wild (AG × AA) 0.22 (23)	Mut × Wild (GG × AA) 0.09 (10)	
ASM (d)	160.56 ± 1.11	156.42 ± 3.29	160.18 ± 1.86	164.0 ± 2.76	0.37
EW at ASM (g)	26.27 ± 0.18 ^b^	25.85 ± 0.56 ^b^	26.09 ± 0.31 ^b^	27.4 ± 0.47 ^a^	0.03
BW at ASM (g)	1731.06 ± 20.81	1757.85 ± 61.92	1763.36 ± 34.93	1834.1 ± 51.81	0.31
BW at 40 Wks (g)	2008.54 ± 33.73	2075.0 ± 100.38	2027.14 ± 56.62	2093.5 ± 83.98	0.77
EW at 40 Wks (g)	46.56 ± 0.12 ^b^	46.49 ± 0.34 ^ab^	47.07 ± 0.19 ^a^	47.33 ± 0.29 ^a^	0.03
EN at 130 to 160 d	2.17 ± 0.48	4.43 ± 1.42	3.0 ± 0.80	2.1 ± 1.19	0.43
EN at 161 to 190 d	16.45 ± 0.42	16.86 ± 1.26	16.27 ± 0.71	14.9 ± 1.05	0.55
EN at 191 to 220 d	16.66 ± 0.28	16.43 ± 0.84	16.95 ± 0.48	16.5 ± 0.70	0.92
EN at 221 to 250 d	15.05 ± 0.28	15.0 ± 0.84	15.36 ± 0.47	14.7 ± 0.70	0.88
EN at 251 to 280 d	11.56 ± 0.37 ^a^	8.57 ± 1.09 ^b^	10.32 ± 0.62 ^ab^	10.5 ± 0.92 ^ab^	0.01
EN at 281 to 310 d	13.08 ± 0.23 ^b^	14.57 ± 0.69 ^a^	12.45 ± 0.39 ^b^	12.9 ± 0.58 ^ab^	0.04

^ab^ Means with different superscripts within the same row differ significantly, Significant: *p* < 0.05, Non-significant: *p* > 0.05, Het: Heterozygous, Mut: Mutant, ASM: age at sexual maturity, EN: egg number/month, EW: egg weight, BW: body weight, Wks: weeks, HW: hatched weight.

## Data Availability

The original contributions presented in the study are included in the article, further inquiries can be directed to the corresponding author.

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
