# Peer review of "Genetic Analysis of HSP70 and HSF3 Polymorphisms and Their Associations with the Egg Production Traits of Bangladeshi Hilly Chickens"

_animals, 2024, doi:10.3390/ani14243552_

Round 1
Reviewer 1 Report
Comments and Suggestions for Authors
The research undertaken by the authors of the manuscript is interesting with some aspects of novelty and follows the current research trend in this field. I believe that the experiment was properly carried out using appropriate material and research methods. In my opinion, some elements require refinement or clarification. In other cases, verification and correction are necessary.
Abstract
Line 33 – I think the word “higher” should be removed.
Introduction
Gene symbols should be italicized.
Line 64 – The term “phenotypic performance” seems unclear.
Lines 80-83 – Information from lines 52-55 is repeated.
Lines 105-107 – Information from the previous sentence (lines 103-104) is repeated.
Materials and Methods
Figure 1 – The title of the figure needs editing.
Line 180 - “The PCR conditions and mixtures are detailed above.” – the notation needs to be clarified.
Line 197 – The sentence needs editing.
Results
Lines 205-205 – It seems that the reference to Table 4 is unnecessary.
Figure 3D – Genotype designation as GG in sample 4 requires verification.
Lines 251-258 – Data contained in Table 4 have been repeated; the description could be simplified.
Lines 262-263 – Frequency values ​​should be written in one way.
Lines 261 – Reference should have been made to Table 2.
Table 5 – I believe that the table should be placed earlier in the manuscript; there is no reference in the text to the content contained in the table.
Lines 280-297 – Please verify the description of the results, because in two cases it seems to be inconsistent with the data in Table 7.
Lines 287 and 311, 353, 374 – Please verify the correctness of the notations “uncommon” and “no common”; it seems that they should be written “different”.
Lines 303-307 – The description of the results requires further clarification in accordance with the data in Table 8.
Figure 4B – Genotype designation as AA in samples 2, 3, 4 and 5 requires verification.
Lines 356-357 – Please verify the correctness of the notation “for this SNP”
Tables 10 and 11 could be presented in reverse order.
Table 13 – Including a group of 2 individuals in the statistical analysis may raise doubts regarding the reliability of the obtained results
Lines 398-400 – The description requires clarification.
Furthermore, the analyzed traits are referred to as production or reproduction – please unify.
Kind regards
Author Response
Thank you for dedicating your time and expertise to reviewing our manuscript. We greatly value the constructive feedback provided, which has helped us refine and improve the quality of our work. Please find detailed responses to each comment below, along with the corresponding revisions and corrections highlighted in the resubmitted manuscript.
A point-by-point response to Comments and Suggestions for Authors
Comments 1: [Line 33 – I think the word “higher” should be removed.]
Response 1: [I have deleted the word “higher”]. I appreciate your observation. I concur with this statement. Consequently, I have amended the sentence in the manuscript's primary text. The modification is located on page one, in the abstract section, specifically line 35, following the word "with" and preceding "egg" highlighted in red of the updated text ["significantly associated with egg numbers (ENs)"].
Comments 2: [Gene symbols should be “italicized”.]
Response 2: I greatly appreciate your comprehensive review of the manuscript. We concur with your observations. I/We have updated the italics [Gene symbols] to underscore this point. Consequently, I/we have amended the Gene symbols in the manuscript's main text. The amendment is located on page two, within the introductory section, notably lines 58-88, highlighted in red in the revised text.
Comments 3: [Line 64 – The term “phenotypic performance” seems unclear.]
Response 3: Thank you so much for thoroughly evaluating the manuscript. I have elucidated the concept of "phenotypic performance" to emphasize this point. Consequently, I have explicitly revised the phenotypic traits, including heat tolerance, weaning weight, milk production, fertility, and disease susceptibility in large animals inside the manuscript's main body. The alteration could be found on page two, in the opening section, specifically in lines 64-65, following the word "traits," marked in red of the revised document.
Comments 4: [Lines 80-83 – Information from lines 52-55 is repeated.]
Response 4: We highly value your comprehensive evaluation of the manuscript. Your observations have been recognized and accepted. The sentence has been modified to emphasize this point. The specified section in the manuscript's introduction has been updated accordingly. The revised text is on page 2 in the introduction section, specifically in lines 79–84, highlighted in red.
Comments 5: [Lines 105-107 – Information from the previous sentence (lines 103-104) is repeated]
Response 5: We much appreciate your thorough assessment of the text. Your observations have been acknowledged and accepted. The phrase has been altered to underscore this point. The designated portion in the manuscript's introduction has been revised appropriately. The amended content is to be found on pages 2-3 inside the introductory section, notably in lines 85–92, marked in red.
Comments 6: [Figure 1 – The title of the figure needs editing.]
Response 6: I appreciate your thorough evaluation of the manuscript. The "Figure 1" title has been revised to emphasize this point. The modification is located on page three, within the Materials and Methods section, specifically at line 108, highlighted in red in the updated manuscript [Figure 1. Location of the conducted study].
Comments 7: [Line 180 - “The PCR conditions and mixtures are detailed above.” – the notation needs to be clarified.]
Response 7: Your observation is appreciated. This statement is agreeable. I have revised the sentence in the main text of the manuscript. The modification appears on page five, within the Materials and Methods section, specifically on line 163, immediately after the word "above," which is highlighted in red in the updated manuscript.
Comments 8: [Line 197 – The sentence needs editing.]
Response 8: Your extensive review of the manuscript is highly appreciated. Your observations are acknowledged and accepted. The sentence has been revised to highlight this point. As a result, we have revised the specified sentence in the main text of the manuscript. The amendment can be found on page five in the Materials and Methods section, specifically in lines 179–180, marked in red of the revised text.
Comments 9: [Lines 205-205 – It seems that the reference to Table 4 is unnecessary.]
Response 9: Your observation is acknowledged. This statement is acceptable. The sentence in the main text of the manuscript has been revised. The modification is located on page six, in the results section, specifically on lines 188-189, highlighted in red in the updated manuscript.
Comments 10: [Figure 3D – Genotype designation as GG in sample 4 requires verification.]
Response 10: Thank you very much for thoroughly evaluating the manuscript. Concerning Figure 3D-genotype designation of sample 4, I agree that it is critical to ensure the genotype data is correct, particularly the GG designation, to reach a firm conclusion. To confirm the results, we verified them three times in our lab and spoke with a lab member and a supervisor before submitting the manuscript. Furthermore, at present, I am outside the affiliated lab in Japan, so it is limited access to verify again at an affiliated lab in Japan.
Comments 11: [Lines 251-258 – Data contained in Table 4 have been repeated; the description could be simplified.]
Response 11: The comprehensive evaluation of the manuscript is highly valued. Your observations have been recognized and accepted. The sentence has been modified to emphasize this point. The specified section in the manuscript's main text has been revised accordingly. The revised text is located on page eight in the results section, specifically in lines 230–240, highlighted in red in the re-edited manuscript.
Comments 12: [Lines 262-263 – Frequency values ​​should be written in one way.]
Response 12: Your comprehensive assessment of the manuscript is appreciated. The frequency values have been updated to highlight this point. The modification appears on page eight in the results section, specifically on lines 240-241, marked in red in the updated manuscript.
Comments 13: [Lines 261 – Reference should have been made to Table 2.]
Response 13: Your detailed evaluation of the manuscript is valued. The reference table has been revised to emphasize this point. The modification is located on page eight in the results section, specifically at line 238, highlighted in red in the updated manuscript.
Comments 14: [Table 5 – I believe that the table should be placed earlier in the manuscript; there is no reference in the text to the content contained in the table.]
Response 14: The thorough assessment of the manuscript is greatly appreciated. Your observations have been acknowledged and accepted. The sequence of Table 5 has been relocated to Table 4, and conversely, Table 4 has been shifted to Table 5 to underscore this point. The relevant section in the manuscript has been revised accordingly. The updated version can be found on page eight within the results section, particularly in lines 227-243, marked in red in the updated version of the manuscript.
Comments 15: [Lines 280-297 – Please verify the description of the results, because in two cases it seems to be inconsistent with the data in Table 7.]
Response 15: The thorough assessment of the manuscript is greatly appreciated. Your observations have been acknowledged and accepted. The specified section has been confirmed and revised to highlight this point. The designated section in the manuscript's main text has been revised accordingly. The revised text can be found on page nine within the results section, specifically on lines 273-275, marked in red in the updated manuscript.
Comments 16: [Lines 287 and 311, 353, 374 – Please verify the correctness of the notations “uncommon” and “no common”; it seems that they should be written “different”.]
Response 16: Your thorough assessment of the manuscript is appreciated. The referenced sections have been revised to highlight this point. The modification can be found on pages 9-12 in the results section, specifically on lines 266, 290, 341, and 353, which are highlighted in red in the updated manuscript.
Comments 17: [Lines 303-307 – The description of the results requires further clarification in accordance with the data in Table 8.]
Response 17: We appreciate your comprehensive assessment of the manuscript. The reference section has been amended to emphasize this aspect. The alteration may be found on pages 9-10 in the results section, particularly on lines 278-286, marked in red in the revised version.
Comments 18: [Figure 4B – Genotype designation as AA in samples 2, 3, 4, and 5 requires verification.]
Response 18: An extensive evaluation of the manuscript would be highly valued. Your observations have been recognized and accepted. The genotype designation AA for samples 2, 3, 4, and 5, as indicated in Figure 4B, has been verified for accuracy. The “faint” or “shadow” bands that appeared along with the expected bands might be due to gel artifacts or uneven gel polymerization/overloading too much DNA in the wells. We ensure the accuracy of the submitted gel image in the manuscript. We verified it 2 to 3 times before submitting the manuscript. However, we also have another image for your kind information. We welcome any suggestions concerning this image.
Comments 19: [Lines 356-357 – Please verify the correctness of the notation “for this SNP”.]
Response 19: Your thorough evaluation of the manuscript is appreciated. The reference SNP has been annotated to highlight this aspect. The correction is located on page 11 in the results section, specifically on lines 336 and 338, highlighted in red in the revised version.
Comments 20: [Tables 10 and 11 could be presented in reverse order.]
Response 20: The comprehensive evaluation of the manuscript is highly valued. Your observations have been recognized and accepted. The sequence previously found in Table 10 has been moved to Table 11, while Table 11 has been reassigned to Table 10 to emphasize this point. The manuscript has been revised in the relevant section as needed. The revised version is located on page 11 in the results section, specifically in lines 330 and 340, highlighted in red.
Comments 21: [Table 13 – Including a group of 2 individuals in the statistical analysis may raise doubts regarding the reliability of the obtained results.]
Response 21: Thank you for thoroughly evaluating the manuscript; it is greatly appreciated. Your observations are acknowledged with respect. In addressing this issue, we utilized only the breeding stock for which phenotypic data was available despite the limited quantity of such stock. Due to the limited availability of experimental birds, we aimed to report the combined genotypic effect of two significant SNPs. This result was included as a novel report on Bangladeshi Hilly chickens.
Comments 22: [Lines 398-400 – The description requires clarification.]
Response 22: Your thorough evaluation of the manuscript is valued. Your observations have been acknowledged and accepted. The specified section has been revised to emphasize this point. The modification is located on page 13 in the results section, specifically between lines 375-378, highlighted in red in the revised manuscript. Furthermore, the revised manuscripts specified the analyzed traits as productive traits.

Reviewer 2 Report
Comments and Suggestions for Authors
1. The Title is “Title...”, The word “Title” should not be part of the title.
2. It is necessary to clarify whether “HSP70” and “HSF3” require italics. There are both italics and non-italics in the text.
3. Line 20: “Hilly chickens” should be written in full name.
4. Line 39: Dot repetition.
5. Line 48: Don't quite understand what “nondescriptive Deshi chickens” means.
6. The Line 175 identifier appears in Table 2.
7. Line 196: “I” should be i.
8. Line 238-244, 251-259 and Table 4, P-value indicate the significant difference of which part of data, Genotype frequency or Allele frequency. There are differences between expression and table content.
9. Line 259-263: “Based on the genotyping data from the five SNPs, haplotypes were constructed and 15 distinct haplotypes (H1 to H15) were identified within the study population as shown in Table 5. The most common haplotype was H1, with a frequency of 34 %, while the frequencies of the other haplotypes ranged from 0.019 to 11.70 %.” I don't know how I came to this conclusion from Table 5, but it should be the information shown in Table 2.
10. Line 271-272: “Notably, G-399A and A258G were linked to egg production during specific intervals”, there should still be C1431A.
11. When describing the results, it should be indicated from which chart the results are obtained. The results obtained from different charts are written together but not indicated, which is easy to cause misunderstanding.
12. Line 292-294: “The SNP C276G showed a significant (P<0.05) association with the EN at 281–310 days of age, and a notably higher EN (P<0.05) was observed in hens with the GG genotype compared with those with the CC and CG genotypes.” The EN of GG genotype hens was not significantly higher than that of CG genotype hens.
13. Line 342-344: “Among these haplotypes, H1 was the most frequently observed with a frequency of 85 %, while H2 was observed in only 15 % of the individuals.” How are these frequencies calculated?
14. The format of some paragraphs and headings is different from others, and the font size of the chart title and comments should be different according to the word template of Animals.
15. There are some blank lines in the text, which will separate the same chart content,
suggested deletion.
16. Line 413-416: According to the content, this sentence should be moved to the next paragraph.
17. In this paper, it is suggested to unify the names of experimental animals, “female Hilly chickens” or “birds” or “Hilly hens”?
Comments on the Quality of English LanguageThe English could be improved to more clearly express the research.
Author Response
Thank you for dedicating your time and expertise to reviewing our manuscript. We greatly value the constructive feedback provided, which has helped us refine and improve the quality of our work. Please find detailed responses to each comment below, along with the corresponding revisions and corrections highlighted in the resubmitted manuscript.
A point-by-point response to Comments and Suggestions for Authors
Comments 1: [The Title is “Title...”, The word “Title” should not be part of the title.]
Response 1: I appreciate your observation. I concur with this statement. Consequently, I have deleted the word “Title” from the title of the manuscript. The modification is located on page one, in the title section, specifically line 2, preceding "Genetic," highlighted in red in the updated manuscript [Genetic Analysis of HSP70 and HSF3 Polymorphisms and their Associations with the Egg Production Traits of Bangladeshi Hilly Chickens].
Comment 2: [It is necessary to clarify whether “HSP70” and “HSF3” require italics. There are both italics and non-italics in the text.]
Response 2: I greatly appreciate your comprehensive review of the manuscript. We concur with your observations. We have updated the italics “Gene symbols” to underscore this point. Consequently, we have amended the Gene symbols in the manuscript's main text; when indicated as a gene, we used italics symbols for both genes. The amendment is located on page two, within the introductory section, notably lines 58-88, highlighted in red in the revised text.
Comment 3: [Line 20: “Hilly chickens” should be written in full name.]
Response 3: Thank you so much for thoroughly evaluating the manuscript. I have elucidated the concept of "hilly chicken" to emphasize this point. Consequently, I have confirmed the full name of this indigenous chicken as "hilly.”. This is an indigenous type of chicken widely known as Hilly (HI) in Bangladesh. Marked in red on page 1, specifically line 20 in the updated manuscript.
Comment 4:[ Line 39: Dot repetition.]
Response 4: Your thorough review of the manuscript is greatly appreciated. Your observations are acknowledged and accepted. The sentence has been revised to highlight this point. As a result, we have revised the specified sentence in the main text of the manuscript. The amendment is to be found on page 1 in the Abstract section, specifically in line 40, following the word “environments” marked in red in the revised text.
Comment 5: [Line 48: Don't quite understand what “nondescriptive Deshi chickens” means.]
Response 5: We appreciate your thorough assessment of the manuscript. Your observations have been acknowledged and accepted. For your information, I am sharing with you that several Indigenous chickens in Bangladesh, such as Naked Neck (NN), nondescriptive Deshi (ND), also known as common Deshi (CD), are farming as egg-type chickens, and hilly (HI) is used for dual purposes. However, we edited the sentence, and the modification is to be seen on page 2, specifically line 49, in the revised manuscript.
Comment 6: [The Line 175 identifier appears in Table 2.]
Response 6: Your detailed evaluation of the manuscript is valued. The reference table has been revised to emphasize this point. The modification is located on page 4 in the Materials and Methods section, specifically at line 157, highlighted in red in the updated manuscript.
Comment 7: [Line 196: “I” should be i.]
Response 7: Your comprehensive assessment of the manuscript is appreciated. The mentioned letter has been updated to highlight this point. The modification appears on page 5 in the Materials and Methods section, specifically on line 178, marked in red in the updated manuscript.
Comment 8: [Line 238-244, 251-259 and Table 4, P-value indicate the significant difference of which part of data, Genotype frequency or Allele frequency. There are differences between expression and table content.]
Response 8: Thank you for thoroughly evaluating the manuscript; it is greatly appreciated. Your observations are acknowledged with respect. In addressing this issue, we utilized only the allele frequency, although stated in the description of the results. For your kind consideration, we reordered the table placement; table 5 was placed in 4 positions, whereas table 4 was moved to table 5 and updated the manuscript accordingly. The amendment appears on page 8 in the Results section, specifically on lines 244-245, marked in red in the updated manuscript.
Comment 9:[ Line 259-263: “Based on the genotyping data from the five SNPs, haplotypes were constructed and 15 distinct haplotypes (H1 to H15) were identified within the study population as shown in Table 5. The most common haplotype was H1, with a frequency of 34 %, while the frequencies of the other haplotypes ranged from 0.019 to 11.70 %.” I don't know how I came to this conclusion from Table 5, but it should be the information shown in Table 2.]
Response 9: Your detailed evaluation of the manuscript is valued. The reference table has been revised to emphasize this point. The modification is located on page 8 in the results section, specifically on lines 238-241, highlighted in red in the updated manuscript.
Comment 10: [Line 271-272: “Notably, G-399A and A258G were linked to egg production during specific intervals”, there should still be C1431A.]
Response 10: The thorough assessment of the manuscript is greatly appreciated. Your observations have been acknowledged and accepted. The specified section has been re-edited and revised to highlight this point. The SNP C1431A has a significant association with higher egg numbers (EN) at 191-220 days in the CA genotype. However, the designated section in the manuscript's main text has been revised accordingly. The revised text is to be found on page 9 within the results section, specifically on lines 273-275, marked in red in the revised manuscript.
Comment 11: [When describing the results, it should be indicated from which chart the results are obtained. [The results obtained from different charts are written together but not indicated, which is easy to cause misunderstanding.]
Response 11: The thorough assessment of the manuscript is greatly appreciated. Your observations have been acknowledged and accepted. The specified section has been revised to highlight this point. The designated section in the manuscript's main text has been revised accordingly.
Comment 12: [Line 292-294: “The SNP C276G showed a significant (P<0.05) association with the EN at 281–310 days of age, and a notably higher EN (P<0.05) was observed in hens with the GG genotype compared with those with the CC and CG genotypes.” The EN of GG genotype hens was not significantly higher than that of CG genotype hens.]
Response 12: Your thorough review of the manuscript is greatly appreciated. Your observations are acknowledged and accepted. The sentence has been revised to statistically higher rather than significantly to highlight this point. As a result, we have revised the specified sentence in the main text of the manuscript. The amendment is to be found on page 9 in the Results section, specifically in lines 270-273, marked in red in the revised text.
Comment 13: [Line 342-344: “Among these haplotypes, H1 was the most frequently observed with a frequency of 85%, while H2 was observed in only 15% of the individuals.” How are these frequencies calculated?
Response 13: Your thorough evaluation of the manuscript is valued. Your observations have been acknowledged and accepted. The specified section has been re-edited to reduce the ambiguity of the reader regarding the findings of the experiment. We deleted this information from this section because another section also stated this issue. We calculated/analyzed the haplotype frequency by “ Direct counting methods” in which the frequency of each haplotype is calculated by dividing the number of occurrences of that haplotype by the total number of haplotypes. The modification is located on page 11 in the results section, specifically between lines 319-322, highlighted in red in the revised manuscript.
Comment 14: [The format of some paragraphs and headings is different from others, and the font size of the chart title and comments should be different according to the word template of Animals.]
Response 14: The thorough assessment of the manuscript is greatly appreciated. Your observations have been acknowledged and accepted. The specified section has been confirmed and revised to highlight this point. The designated section in the manuscript's main text has been revised accordingly. The revised text is to be found specifically marked in red in the updated version.
Comment 15: [There are some blank lines in the text, which will separate the same chart content; suggested deletion.]
Response 15: The thorough assessment of the manuscript is greatly appreciated. Your observations have been acknowledged and accepted. The specified section has been confirmed and revised to highlight this point. The designated section in the manuscript's main text has been revised accordingly. The revised text is to be found specifically marked in red on the updated manuscript.
Comment 16: [Line 413-416: According to the content, this sentence should be moved to the next paragraph.]
Response 16: The thorough assessment of the manuscript is greatly appreciated. Your observations have been acknowledged and accepted. The specified section has been moved and revised to highlight this point. The designated section in the manuscript's main text has been revised accordingly. The revised text is to be found on page 13, specifically in lines 391-392 marked in red in the re-edited version.
Comment 17: [In this paper, it is suggested to unify the names of experimental animals, “female Hilly chickens” or “birds” or “Hilly hens”?]
Response 17: Thank you so much for thoroughly evaluating the manuscript. I have elucidated the concept of "hilly chicken" to emphasize this point. Consequently, I have confirmed the full name of this indigenous chicken as “Hilly chicken”. This is an indigenous type of chicken widely known as hilly (HI) in Bangladesh.
[The English could be improved to more clearly express the research.]
Response: We sincerely appreciate your thorough assessment of the manuscript. Your observations have been carefully considered and accepted. To enhance the quality of the manuscript, we have had it professionally edited by an English language expert.

Reviewer 3 Report
Comments and Suggestions for Authors
The topic of the research is important for the development of the region and is aimed at preserving local chickens. However, there are many shortcomings in the work done.
1. It is desirable to consider in more detail the possible influence of the HSP70 gene on the productivity of animals in the introduction.
2. The associations of the genotype with the productive qualities of birds were made on too small a population. It is necessary to increase the population, or provide data on productivity in the appendix.
3. How many birds were sequenced and genotyped?
4. Was the calculation of linkage disequilibrium between polymorphisms carried out? It is desirable to take into account LD, since we are talking about haplotypes.
Author Response
Thank you for dedicating your time and expertise to reviewing our manuscript. We greatly value the constructive feedback provided, which has helped us refine and improve the quality of our work. Please find detailed responses to each comment below, along with the corresponding revisions and corrections highlighted in the resubmitted manuscript.
A point-by-point response to Comments and Suggestions for Authors
Comment 1: [It is desirable to consider in more detail the possible influence of the HSP70 gene on the productivity of animals in the introduction.]
Response 1: Your thorough review of the manuscript is greatly appreciated. Your observations are acknowledged and accepted. The sentence has been revised in some parts of the introduction to highlight this point. As a result, we have revised the specified sentence in the main text of the manuscript. The amendment is to be found on pages 2-3 in the Introduction section, specifically in lines 58-92, following the word red marked in the revised text.
Comment 2. [The associations of the genotype with the productive qualities of birds were made on too small a population.]. It is necessary to increase the population or provide data on productivity in the appendix.]
Response 2: Thank you for thoroughly evaluating the manuscript; it is greatly appreciated. Your observations are acknowledged with respect. In addressing this issue, we utilized only the breeding stock for which phenotypic data was available, despite the limited quantity of such stock. Due to the limited availability of experimental birds, we aimed to report the association between the SNPs in HSP70 and HSF3 with egg production traits of Hilly chickens in Bangladesh. This result was included as a novel report on Bangladeshi Hilly chickens.
Comment 3. [How many birds were sequenced and genotyped?]
Response 3: Your observation is acknowledged. This statement is acceptable. In our study, we utilized 150 mature female hilly chickens for sequencing and genotyping.
Comment 4. [Was the calculation of linkage disequilibrium between polymorphisms carried out? It is desirable to take into account LD since we are talking about haplotypes.]
Response 4: Thank you for your insightful input. We initially focused on haplotype analysis because of the limited sample size (n=150), which may need more power for a comprehensive LD study. Performing LD analysis in a limited group may provide unreliable results that fail to adequately represent the linkage structure of the larger population. Given these constraints, we focused on direct haplotype analysis to tackle our principal study issue on egg production traits.
Furthermore, because haplotype analysis may successfully identify connections in small sample sizes, this method captures the key genetic links for our research goals without the additional complication of perhaps inaccurate LD measures. Thank you for your awareness of the limitations encountered in our research. We hope that this makes our approach clearer.

Round 2
Reviewer 1 Report
Comments and Suggestions for Authors
Thank you for taking into account the comments and suggestions, but the manuscript still needs improvement. Please check the gene symbols carefully (not all of them are written in italics). On line 70, the protein symbol is written in italics (in this case, we do not use italics). The frequency notation on line 241 has still not been standardized. Three decimal places are included, while previously values ​​are given with two decimal places. Association analysis still requires verification. With such a superscript notation, the description of the results seems to be incompatible with the data presented in the tables. An example is the sentence on lines 261-263. According to Table 7, a significantly higher EW (P < 0.05) was observed in birds with the mutant GG genotype compared with birds with the AA genotype; not with AG. In the sentence on lines 269-270, it can be assumed that the AA genotype hens produced more eggs than the AG and GG genotype hens. However, the values ​​for genotypes AA and AG have the same superscript, and the table legend states "Means with different superscripts within the same raw differ significantly." Therefore, it is necessary to carefully review the manuscript for a way to indicate the significance of differences and to describe the relationships between genotypes, haplotypes and the analyzed traits.
Best Regards
Author Response
Dear Reviewer,
We appreciate your review of our manuscript and your valuable comments and suggestions. Your constructive feedback is appreciated and has contributed to the enhancement of the quality and clarity of our work. We present a comprehensive response to your comments and summarise the revisions made.
Point-by-point Response to Comments and Suggestions for Authors
Comment 1: [ Gene symbols must be italicized in the manuscript.]
Response 1: I appreciate your observation. We have reviewed the entire manuscript carefully and ensured consistent italicisation of all gene symbols (e.g., HSP70, HSF3). Protein names (e.g., HSP70, HSF3) and family names (e.g., HSPs, HSFs) are not italicised, in accordance with established formatting conventions. The amendment is located on page two, within the introductory section, notably lines 58, 66, 68, 71-72, 82-83, and 85, highlighted in yellow in the revised manuscript.
Comment 2: [The frequency notation on line 241 has still not been standardised].
Response 2: Your thorough evaluation of the manuscript is valued. The specified section, particularly the frequency values, has been revised to ensure consistency; specifically, two decimal places were employed and this aspect has been emphasized. The modification is located on page 8 in the results section, specifically on line 245, and is highlighted in yellow in the updated manuscript.
Comment 3: [Association analysis still requires verification. With such a superscript notation, the description of the results seems to be incompatible with the data presented in the tables. An example is the sentence on lines 261–263. According to Table 7, a significantly higher EW (P < 0.05) was observed in birds with the mutant GG genotype compared with birds with the AA genotype; not with AG.]
Response 3: Thank you for highlighting this. Upon review, we found that the sentence in lines 261–263 was indeed inconsistent with the data presented in Table 7. We have corrected the text and table content to accurately reflect the statistical differences indicated in the table. Specifically, the revised sentence now reads:
"The A-68G SNP demonstrated a significant association with egg weight (EW) at 40 weeks of age, with birds possessing the mutant GG genotype exhibiting a notably higher EW (P < 0.05) in comparison to those with the wild AA genotype (Table 7)".
The amendment is located on page 9 in the results section, specifically on lines 267-270, and is highlighted in yellow in the updated manuscript.
This revision ensures alignment between the description in the text and the data in Table 7.
Comment 4: "In the sentence on lines 269–270, it can be assumed that the AA genotype hens produced more eggs than the AG and GG genotype hens. However, the values for genotypes AA and AG have the same superscript, and the table legend states, "Means with different superscripts within the same row differ significantly."
Response 4: Your observation regarding this discrepancy is indeed accurate. The statistical analysis and superscript notations in Table 7 were re-evaluated to confirm their accuracy. The superscript notations in Table 7 remained, and the table legend remained the same. The sentence on lines 269–270 has been revised to accurately depict the relationships between genotypes and egg production traits as follows:
"In addition, the AA genotype hens produced the highest number of eggs (16.00), which was significantly greater than the other two genotypes; however, no significant variations were observed between the AG and GG genotypes.".
The updated description is now consistent with the amended table data and the conventions for superscript notation and is to be found on page 9 in the results section, specifically on lines 276-279, and is highlighted in yellow in the updated manuscript
Comment 5: [It is necessary to carefully review the manuscript for a way to indicate the significance of differences and to describe the relationship between genotypes, haplotypes, and the analyzed traits].
Response 5: A comprehensive review of the manuscript has been conducted, with particular emphasis on the consistency of superscript notations, the reporting of statistical significance, and the relationships among genotypes, haplotypes, and the analyzed traits. The tables and their corresponding text descriptions have undergone meticulous cross-verification to ensure precision. We have clarified the methodology for statistical comparisons in the Materials and Methods section to enhance transparency. The revisions enhance the clarity and accuracy of the results and their interpretation.
Your valuable insights and suggestions have helped us improve the manuscript and enhance its clarity and quality.

Reviewer 3 Report
Comments and Suggestions for Authors
The authors were able to correct the main comments or provide answers regarding the manuscript. I recommend accepting the manuscript for publication.
Author Response
Dear Reviewer,
We sincerely thank you for your thoughtful and encouraging feedback on our manuscript [Genetic Analysis of HSP70 and HSF3 Polymorphisms and Their Association with Egg Production Traits of Bangladeshi Hilly Chickens and ID Animals—3281984]. We greatly appreciate the time and effort you dedicated to reviewing our work and your recommendation for its acceptance. Your valuable insights and suggestions have helped us improve the manuscript and enhance its clarity and quality.
We are grateful for your support and encouragement, and we look forward to seeing our work published.
Thank you once again for your kind words and constructive evaluation.
Sincerely,
Md.Yousuf Ali
Bangladesh Livestock Research Institute
